# Spin Lattice (T1) and Magnetization Transfer Saturation (MT_sat_) Imaging to Monitor Age-Related Differences in Skeletal Muscle Tissue

**DOI:** 10.3390/diagnostics12030584

**Published:** 2022-02-24

**Authors:** John Cameron White, Shantanu Sinha, Usha Sinha

**Affiliations:** 1Department of Physics, San Diego State University, San Diego, CA 92182, USA; jocwhite@me.com; 2Muscle Imaging and Modeling Laboratory, Department of Radiology, University of California San Diego, San Diego, CA 92093, USA; shsinha@health.ucsd.edu

**Keywords:** normal age-related changes in muscle tissue composition, T1 imaging, magnetization transfer saturation imaging, macromolecular fraction, inflammation

## Abstract

Background: The aim was to compare spin-lattice relaxation (T1) mapping from sequences with no fat suppression and three fat suppression methods and Magnetization Transfer Saturation (MT_sat_) mapping, to identify regional and age-related differences in calf muscle. These differences may be of clinical significance in age-related loss of muscle force. Methods: Ten young and seven senior subjects were imaged on a 3T MRI scanner using a 3D Fast Low Angle Shot sequence without and with different fat suppression and with MT saturation pulse. Bland–Altman plots were used to assess T1 maps using the fat unsuppressed sequence as the reference image. Age and regional differences in T1 and in MT_sat_ were assessed using two-way factorial analyses of variance (ANOVAs) with Bonferroni-adjusted independent sample t-tests for post hoc analyses. Results: A significant age-related increase in T1 and decrease in MT_sat_ was seen in the calf muscles. The largest size effect was observed in the T1 sequence with fat saturation. Conclusions: T1 increase with age may reflect increase in inflammatory processes while the decrease in MTsat may indicate that magnetization transfer may also be associated with muscle fiber macromolecules. T1 and MT_sat_ maps of calf muscle have the potential to detect regional and age-related compositional differences in calf muscle.

## 1. Introduction

It is a well-known fact that as we age, muscle mass decreases even in the absence of disease [1,2]. Beyond the loss of muscle mass, there is a disproportionately greater loss of muscle strength as we age, termed dynopenia [3]. This disproportionate loss translates to a loss of specific force defined as the force relative to the cross-sectional area. Dynopenia is a multifactorial disease and current research effort has been directed to studying muscular and neural determinants of loss of specific force. Increasingly, the contribution of the extracellular matrix (ECM) to dynopenia is being recognized with initial evidence coming from animal studies. It has been identified in rodent models that muscle weakness also arises from the ECM impairing the muscle’s capacity to transmit force [4]. ECM remodeling with age is manifested as an increase in intramuscular connective tissue and in collagen content [5]. These changes could potentially contribute to aging related loss of muscle force [4]. Currently, there are no non-invasive techniques to monitor in vivo changes in collagen content.

Collagen is a large macromolecule and protons bound to collagen cannot be detected directly using MRI due to the very low T2 values (<0.01 ms), which cause a very rapid decay in the signal, preventing its observation. Magnetization transfer contrast (MTC) MRI provides an indirect observation to detect macromolecules such as myelin in tissue [6,7]. MTC can be studied by different methods ranging from the simple magnetization transfer ratio (MTR) to the complicated quantitative magnetization transfer imaging (qMT) that permits the extraction of the macromolecular fraction (*f*); qMT involves longer acquisition times and complex computation [8,9]. In contrast, the semi-quantitative magnetization transfer saturation contrast imaging, termed MT_sat_ mapping, provides a robust and fast (~6 min acquisition) clinical alternative to the more involved qMT mapping [10].

The aging muscle tissue exhibits a chronic low-grade inflammatory process that has been termed “inflammaging” [11,12]. The chronic inflammation of the aging muscle is closely associated with impairment of metabolic pathways that exacerbate inflammation [13]. Metabolism-induced inflammation, termed “metaflammation”, has been shown to potentially trigger obesity-induced insulin resistance. Further, other studies have linked adipose tissue to chronic low-grade systemic inflammation, with adipose tissue triggering increases in inflammatory molecules [14]. The increase in fat mass accompanied by a decrease in lean mass with age may be the underlying cause of age-related chronic low-grade inflammation (“inflammaging”) [15].

Intra- and intermuscular fat infiltration with age in skeletal muscle is well known and has been documented in several studies [16]. The effect of fat infiltration (increase in fat fraction) on T1 relaxation times is a decrease in T1 values since the T1 of fat is lower than that of muscle. On the other hand, inflammation leads to a higher “free water” content that results in elevated T1 in aging muscle compared to normal young healthy muscle. Since compositional changes in the aging muscle include both inflammation and fatty infiltration, T1 mapping may potentially reflect these changes. Further, in addition to being sensitive to fat infiltration and inflammation, T1 is also affected by the macromolecular content. For example, the differences in T1 relaxation between grey and white matter have been attributed to their different myelin content: white matter has a high myelin content (~30%) that is made up of proteins [17]. Protons bound to myelin accelerate the T1 relaxation of the visible proton pool through magnetization transfer. Thus, evaluation of T1 with magnetization transfer may provide information about underlying changes in macromolecular content. For example, fibrosis (intramuscular connective tissue) increases with age [16] and, thus, a T1 measurement using an off-resonance pulse (that induces magnetization transfer) may be a marker of the degree of fibrosis. The measured changes in T1 may thus reflect the combined changes in fat and fibrosis fraction and inflammation in musculoskeletal tissue [18]. In order to disambiguate the effects of inflammation from fat, it is critical to use fat suppression. Alternatively, T1 estimated from a fat unsuppressed sequence may serve as a surrogate marker of fat infiltration in muscle.

The focus of the current study was the application of T1 and MT_sat_ mapping to explore age-related changes in calf muscle. In addition, the study explored the effects of different fat suppression methods on the observed T1 values, including incidental magnetization transfer effects from off-resonance fat saturation pulses.

## 2. Materials and Methods

### 2.1. Subjects

Institutional Review Board (IRB) approval was obtained from San Diego State University’s Human Research Protection Program (HRPP) and all subjects were imaged after obtaining written informed consent. The study included a cohort of 10 young healthy subjects (5M/5F, 24.5 ± 3 years) and seven healthy senior subjects (6M/1F, 65 ± 8 years). The criteria for inclusion were that subjects (young and senior) should be free from neuromuscular disease and be moderately active. Subjects participating in competitive sports or those with any surgical procedure on the lower leg were excluded. The dominant leg was imaged for all subjects.

All subjects were imaged in a Siemens Prisma 3.0 Tesla (Siemens PRISMA, software version: VE11C, Erlangen, Germany). Subjects were positioned feet first, supine in the scanner. The lower leg (posterior portion) was positioned on the Spine 32 coil (12 channels activated) while a 4-channel flex coil was wrapped around the anterior part of the lower leg. All scans spanned the upper region of the calf muscle.

An a priori power analysis (α = 0.05, 1 − β = 0.80) was performed based on the difference in Magnetization Transfer Ratio (MTR) observed between young and older human participants in the age range of 20 to 70 years old [19]. The results suggest that a minimum of 6 subjects from each age group would be required to detect between-group differences of similar size (MTR of 33 ± 1.2 at 20 years with a decrease of 5% at a mean age of 65 years). It should be noted that while MTR and MTsat both measure the magnetization transfer effect, MTsat—unlike MTR—is not influenced by B1 inhomogeneities, changes in sequence parameters, or changes in T1.

### 2.2. MRI Acquisition

#### 2.2.1. T1 Mapping

Four acquisitions based on 3D FLASH sequences were performed (TR = 35 ms; TE = 2.86 ms (sequences (i) through (iii) below) or 3.54 ms (sequence (iv) below); matrix = 128 × 128 × 32 (25% slice oversampling); voxel size = 1.4 × 1.4 × 5 mm^3^; GRAPPA = 2 acceleration). All sequences were acquired at three flip angles (5°, 15°, 25°) to calculate T1 after correction of the flip angle using B1+ maps [20]. Each acquisition took 1 min 58 s. The four sequences were acquired with the following fat suppression techniques: (i) no fat suppression: 3D FLASH without any fat suppression; (ii) fat suppression accomplished with a saturation pulse at 440 Hz off resonance (chemical saturation); (iii) water excitation using an on-resonance 1:1 binomial composite pulse; this is denoted “fast”; and (iv) water excitation using an on-resonance 1:2:1 binomial composite pulse; this is denoted “normal”.

The relative B1+ maps to correct the nominal flip angles were generated using a 20 s turboFLASH (TFL) sequence with and without magnetization preparation [20]. The B1+ map was calculated as the ratio of the measured angle to the nominal flip angle. The B1+ scan time was 40 s including adjustment of the coil tuning.

#### 2.2.2. Magnetization Transfer Saturation (MT_sat_) Imaging

Three 3D FLASH acquisitions were used: TR = 50 ms, TE = 4.55 ms, matrix = 128 × 128 × 32 (25% slice oversampling), voxel size 1.4 × 1.4 × 5 and GRAPPA acceleration factor = 2. The water excitation (normal) was used for selective water excitation as this was determined to be more effective for fat suppression than the water excitation (fast). The magnetization transfer pulse was a Gaussian RF pulse, 375 Hz bandwidth, 9.984 ms duration, 1.2 kHz offset, 500° flip angle. Images were acquired at three different flip angles to compute the MT_sat_ map: PD_w_ (4°), MT_w_ (10° with the magnetization transfer pulse), and T1_w_ (20°). The acquisition time for each sequence (PD_w,_ MT_w_, T1_w_) was ~110 s, so that a total of ~6 min was required for MT_sat_ mapping.

### 2.3. Computation of T1 and MT_sat_ Maps

The mean and standard deviation of MT_sat_ and MTR values were determined in ROIs placed in the following muscle compartments of the calf muscle in the middle slice of the acquired volume: medial gastrocnemius (MG), lateral gastrocnemius (LG), soleus (SOL), tibialis posterior (TP), and tibialis anterior (TA). The ROIs for the muscle regions varied in size; for the smaller muscle compartments (TA and TP), ROIs had a size of 70–150 voxels while the ROIs had a size of approximately 200 voxels for the SOL, MG, and LG muscles. 

T1 and MT_sat_ maps were calculated in MATLAB using qMRLab (https://github.com/neuropoly/qMRLab/releases, accessed on 11 January 2018) [21]. A brief summary of the T1 and MTsat computations is included below. The T1 maps were computed using the linearized version of the FLASH equation:(1)$$\frac{S_{FLASH}}{\sin\left(\alpha \right)}=E_{1}\frac{S_{FLASH}}{\tan\left(\alpha \right)}+M_{0}\left(1-E_{1}\right)$$
where *S_FLASH_* is the signal intensity for the image acquired with flip angle *α*, *E*_1_ = exp(−*TR/T*1), and *M*_0_ is the equilibrium magnetization. Images at the three flip angles are used in the *T*1 maps and the nominal flip angle is corrected using the B1 + maps.

The computation of MT_sat_ is detailed in earlier studies [10,22] and is given by:(2)$$\;S_{mt}=A\;\alpha \frac{R_{1}TR}{\frac{\alpha^{2}}{2}+\delta +R_{1}TR}$$
where *S_mt_* is the signal intensity of the MT_w_ sequence, *A* is the amplitude of the spoiled gradient echo at echo time, TE, under fully relaxed conditions (*R*_1_*TR* >> 1, *α*
*= π*/2), *R*_1_ is the spin lattice relaxivity (*R*_1_ = 1*/T*1), *TR* is the repetition time, *α* is the flip angle of the readout pulse for the MT_w_ sequence, and *δ* is the MT saturation term that exceeds the readout pulse effects [10]. To solve for *δ*, Equation (2) requires estimates for *A* and *R*_1_. *A* and *R*_1_ are estimated from the PD_w_ and T1_w_ FLASH sequences at appropriate flip angles that result in predominant proton density (PD) and T1 weighting, respectively [10]. The latter two FLASH sequences are acquired without the MT pulse.

### 2.4. Statistical Analysis

The mean and standard deviation of MT_sat_ and T1 values were determined in ROIs placed in the medial gastrocnemius (MG), lateral gastrocnemius (LG), sol (SOL), tibialis posterior (TP), and tibialis anterior (TA). The outcome variables of the analysis were T1 (with no fat suppression and three fat suppression methods) and MT_sat_. Unless stated otherwise, data are presented as means ± SD. Normality of data was verified visually by histograms and numerically using the Shapiro–Wilk test (*p* > 0.05). In order to evaluate the three fat suppression methods, T1 values measured on the fat suppressed sequences were compared to the fat unsuppressed sequence (reference data) using Bland–Altman plots.

Differences between age and intermuscular regions, in addition to potential interaction effects, were assessed using two-way factorial analyses of variance (ANOVAs) (“age” × “region”). Levene’s test was used to test the assumption of homogeneity of variance and, in the case of significant ANOVA results for the factor “region”, Bonferroni-adjusted independent sample t-tests were used for post hoc analyses.

## 3. Results

### 3.1. Fat Suppression

The ratio of the signal from the regions of interest placed in the subcutaneous fat and muscle was evaluated for the four T1 sequences. This ratios for the unsuppressed, fat saturation, water excitation using a 1:1 and a 1:2:1 pulse sequence for a young/old subject were: 0.93/1.06, 0.21/0.32, 0.19/0.29, and 0.18/0.28, respectively. All three fat suppression sequences provided a similar extent of fat suppression, with the water excitation 1:2:1 resulting in a marginally better fat suppression (lowest value of the ratio of fat to muscle signal intensity).

### 3.2. T1 Mapping

Figure 1 shows representative images for a young subject including the images acquired at the three flip angles, the B1+ map and the T1 maps (color coded for better visualization of spatial patterns), and the ROIs for the evaluation of T1 in different muscles. The acquired data and the computed T1 maps were of similar quality for senior subjects. The relative B1+ map values ranged from 0.8 to 1.2, with higher values in the posterior region. Table 1 is a summary of the T1 values from the four sequences averaged over the young and senior subjects. Significant differences between young and senior subjects (with longer T1s for the senior cohort) were seen in all the fat suppressed sequences. Significant regional differences were seen primarily between the lateral gastrocnemius (longer T1) and the other four muscles (medial gastrocnemius, soleus, anterior tibialis, and posterior tibialis).

Figure 2a is the Bland–Altman plot comparing, for young subjects, T1 from the fat saturated sequence to the reference T1 from the fat unsuppressed sequence. The plot compares the mean of the two measurements (*x*-axis) to the difference in the two measurements (*y*-axis). The mean of the differences (bias, green line) was high at 0.5461 and the ±95% confidence lines were at 0.3956 and at 0.6965 (the range within which 95% of the differences between one method and the other will occur). A large bias and the large 95% range show these two techniques are not equivalent to each other. Figure 2b is the Bland–Altman plot comparing the T1 from the sequence with 1:1 water excitation (water fast) to the reference T1 from the fat unsuppressed sequence. There is only a small bias (0.0268) and a small ±95% confidence range (0.1237 to −0.0701) confirming that these two methods are equivalent to each other. Figure 2c is the Bland–Altman plot comparing the T1 from the sequence with 1:2:1 water excitation to the reference T1 from the fat unsuppressed sequence. As in the 1:1 water excitation sequence, there is only a small bias (0.0251) and a small ±95% confidence range (0.2171 to −0.1652), confirming that these two methods are equivalent to each other.

### 3.3. MT_sat_ Mapping

T1_w_, MT_w_, and PD_w_ images, and computed MT_sat_ images for a senior subject, are shown in Figure 3. Images of similar quality were obtained for the young subject. Table 2 lists the MT_sat_ for the different muscles in young and old subjects. Significant age-related differences in MT_sat_ were found between the young and senior subjects, with lower values of MT_sat_ in the senior cohort. In regional differences, TA was significantly lower than SOL, MG, and TP, while LG was significantly lower than SOL. No significant (age × muscle region) effect was seen. Appendix A lists the ROI values of T1 (four different sequences) and MT_sat_ for all the subjects in the current study.

## 4. Discussion

Fat suppression is critical for quantitative muscle imaging (T1 mapping or MT_sat_) as fat increasingly infiltrates muscle with age and in abnormal conditions such as sarcopenia and dystrophy. The presence of fat can confound both T1 and MT values since fat has a lower T1 and no significant magnetization transfer effects. Fat suppression can be accomplished in several ways and the current study compared three methods: fat saturation (chemical saturation at the fat frequency) and two modes of water excitation using composite pulses. When a RF pulse centered on the fat frequency (420–440 Hz downstream from water frequency) is included prior to the excitation pulse, fat protons are selectively saturated, and if this magnetization is dephased by a spoiler gradient, then the fat protons do not contribute to the image signal. In water excitation (WE), water protons are selectively excited while the fat protons are not excited. This is accomplished by composite RF pulses; the composite RF pulses explored in the current work were 1:1 and 1:2:1 RF pulses for water excitation.

In order to compare T1 computed from the sequences with different fat suppression methods, the fat unsuppressed sequence was used as the reference sequence. This reference was valid since data from young subjects were only used in comparing the T1 values. Younger subjects are known to have low fat fractions (2–3%) and thus it is anticipated that estimates of T1 will not be biased due to the presence of fat; that is, muscle T1 values from the fat unsuppressed sequence will not have significant contributions from fat in young subjects and can be used as a reference sequence in the Bland–Altman (BA) plots. This enables one to determine the effect of the different fat suppression schemes on the measured muscle T1 values. BA plots show that the two water excitation sequences are comparable to the fat unsuppressed sequence. By comparison, T1 estimated from the fat saturated sequence showed the lowest values, with a significant bias of ~44% compared to the reference. The decrease in T1 in the fat saturated sequence can be potentially attributed to incidental magnetization transfer effects from the off-resonance fat saturation pulse applied 440 Hz downstream from the water peak. It is important to realize that off resonance pulses, including the fat saturation pulse, cause magnetization transfer effects, resulting not only in a decrease in the longitudinal magnetization but also a decrease in T1 in the free pool; the lower T1 in the presence of an off-resonance saturation pulse is denoted as T1_sat_ (T1 in the presence of magnetization transfer effects). This is an important observation since the fat saturation (chem-sat) technique is the most commonly used fat suppression technique in quantitative muscle imaging and can inadvertently bias the measurements of T1. The extent of T1 reduction will depend on the specifics of the fat saturation pulse and the strength of magnetization transfer in the tissue.

Significant age-related increases were found in muscle T1 values extracted from the fat-saturated and from the water excitation (fast), while that from water excitation (normal) showed a trend to higher values. The fat saturated T1 mapping sequence showed the biggest average age increase in T1 (3.2% increase) compared to water sat (fast), which also showed significant differences, but the effect size was only a 1.9% increase with age. The larger differences in T1 found in the fat saturated sequence may potentially arise from the fact that this sequence reflects magnetization transfer effects in addition to changes in longitudinal magnetization. By comparison, the water excitation sequence reflects the changes in intrinsic muscle T1. The T1 increase with age measured by this latter sequence may be attributed to muscle inflammation that occurs in elderly subjects [11,12]. In addition to inflammation, changes in muscle type with age can potentially also affect intrinsic T1. Aging atrophy is mainly due to a reduction in the number and size of Type II muscle fibers [23]; this reduction in Type II fiber size with age may lead to larger extracellular spaces with longer T1 values, which leads to an overall increase in T1.

The T1 from the fat saturated sequence reflects both the muscle changes in intrinsic T1 and the T1 differences arising from changes in magnetization transfer effects with age. Since T1 of the fat saturated sequence showed the largest T1 increases with age, this implies that, with age, magnetization transfer effects decreased. This decrease in magnetization transfer with age would then result in a smaller decrease in T1 due to T1_sat_ effects. This is surprising since the prevailing hypothesis is that collagen is the primary macromolecule responsible for the magnetization transfer effect observed in muscle tissue [24,25], and collagen is known to increase with age [26]. Then, it would be anticipated that there are stronger magnetization transfer effects with age and larger, rather than smaller, T1_sat_ effects. It should be noted that no significant differences were found in T1 values extracted from the fat unsuppressed sequence. This latter null finding can be explained based on the fact that fat fraction increases with age and, since fat has a lower T1 than muscle, this results in lower T1 values. In parallel, there is an age-related increase in intrinsic T1 potentially from inflammation and Type II atrophy effects. It is very likely that the absence of age-related differences in T1 computed from the fat unsuppressed sequence arises from the opposing effects of inflammation and fat increase on T1. In this context, it should be noted that an earlier study on T1 mapping reported no age-related changes in calf muscle, but the study did not use fat suppression [27]. In fact, the latter study concluded that the lack of correlation between age and T1 arises from the inability to distinguish water and fat signal contributions. This gave rise to confounding effects from an age-related increase in fat content resulting in lower muscle T1 values, while an increase in water due to a shift in fiber types and/or inflammation led to longer muscle T1 values. The same effect is seen in the fat unsuppressed sequence in the current study. In addition, T1 mapping has been suggested as a surrogate marker for fat in dystrophic muscle [28]; however, it is then important to ensure that the only changes in muscle T1 arise from fat infiltration, and are not confounded by inflammation or other intrinsic factors in muscle.

In regional differences, the lateral gastrocnemius had longer T1 values that were significantly different from those of the soleus, medial gastrocnemius, and posterior tibialis. It is possible that the T1 differences between muscles may arise from the relative proportion of Type I and Type II fibers. The ratios of Type I to Type II fibers in the soleus, gastrocnemius (medial/lateral), and anterior tibialis are approximately 70/30, 50/50, and 75/25, respectively [29,30,31]. The LG and MG have a lower proportion of Type I fibers, which are known to have a smaller diameter and a higher macromolecular fraction than Type II fibers. It is conceivable that the muscles with a greater fraction of Type II fibers exhibit a lower magnetization transfer effect since they have lower macromolecular fractions. Although this is speculative, this may be the reason lateral gastrocnemius has the longest T1 value (smaller T1sat effects).

MT_sat_ of the senior cohort was significantly lower than that of the young cohort. This result is contradictory to the hypothesis that the known increase in collagen with age may result in an increase in magnetization transfer effects and, thus, MT_sat_. However, as also seen from T1 values extracted from the fat suppressed sequences, the results point to a decrease in magnetization transfer effects with age. In a prior study, the Magnetization Transfer Ratio (MTR) evaluated at the mid-thigh and mid-calf regions was also shown to correlate significantly negatively with age [19]; this finding is similar to the results of age-related changes in MT_sat_ in the current paper. This latter study did not use fat suppression but included the fat fraction as a covariate in the regression analysis of MTR to age, and the authors concluded that their results suggest an MTR age-dependence independent of age-related muscle lipid increases, presumably reflecting myofiber quality and density changes. Compared to MT_sat_, MTR values are influenced by changes in T1 values and B1 inhomogeneities. Further, it should be noted that, in muscle, the specific macromolecular pool responsible for the observed magnetization transfer effects has not yet been established. In contrast, in the brain, it has been established that the macromolecular fraction from quantitative magnetization transfer (qMT) has a high degree of correlation to myelin content [32]. Although it has been hypothesized that collagen is the main macromolecule involved in MT in muscle [24,25], this is yet to be confirmed by histological analysis of muscle biopsy samples to identify correlations of collagen content to qMT and MT_sat_ parameters. The results of the current study show that the microstructural/macromolecular changes in muscle tissue that affect age-related differences in magnetization transfer contrast may be more complex than just an increase in collagen content in the extracellular matrix with age. Insight into other mechanisms that could potentially explain the observed MT_sat_ results comes from a rat model study of Magnetization Transfer Ratio (MTR). In the latter study, MTR was used to track muscle fiber formation after injection of human muscle progenitor cells for development of muscle tissue [33]. MTR increased with myogenesis and correlated well with muscle contractility measurements. The authors of the latter paper advanced the hypothesis that higher MT in muscle may arise from a large abundance of macromolecules in the form of aligned muscle fibers in well-developed muscle tissue (resulting in an increase in MT with muscle development) [33]. If the macromolecules responsible for the MT effect are in the form of aligned muscle fibers, then the atrophy of muscle fibers with age may actually lead to a loss of the macromolecular pool, and hence to a decrease in MT_sat_, as seen in the present study. These studies suggest that biopsy studies are critical to show the correlations of MT indices to tissue parameters in muscle. Analysis of biopsy samples can quantify the total amount/type of collagen, fiber type-specific atrophy, muscle fiber diameter, number of muscle fibers, and thickness of the extracellular matrix. Correlation of these parameters to MT_sat_ will help identify the microstructural change (or combinations of changes) that lead to the observed MT_sat_ changes at the tissue level.

The study has several limitations. It has a small sample size as the focus was to evaluate different fat suppression methods for T1 mapping and to establish the feasibility of T1 and MT_sat_ mapping to monitor age-related differences between young and senior subjects. However, even with this sample size, significant differences in T1 and in MT_sat_ between different muscles of the calf, and between young and senior subjects, were seen. Larger studies will be required to study the potential of using T1 and MT_sat_ to characterize the aging muscle. Further, MT_sat_ was not validated using a reference acquisition or by biopsy analysis of muscle tissue; the latter was beyond the scope of this study. However, an earlier phantom study confirmed that MT_sat_ changed linearly with macromolecular fraction. Future studies can extend this by in vivo correlation of the macromolecular pool fraction from qMT studies to MT_sat_.

In conclusion, this study evaluated in the calf muscle (i) three different fat suppression methods and a fat unsuppressed T1 mapping; (ii) four T1 mapping sequences to monitor age-related and regional muscle differences; and (iii) MT_sat_ to monitor age-related and regional differences. The age-related increase in T1 is potentially a marker of inflammation and/or preferential atrophy of Type II fibers. Age-related decrease in MT_sat_ may reflect a combination of changes from ECM remodeling and muscle fiber loss and atrophy. T1 with fat saturation combines additively the changes from intrinsic T1 and from MT_sat_, yielding a sensitive biomarker of age-related changes in skeletal muscle. T1 and MT_sat_ mapping can be accomplished in three and six minutes, respectively, using sequences routinely available on scanners, and can thus be added to any clinical protocol. The potential to monitor age-related changes has been demonstrated in the current exploratory study but needs to be evaluated on a larger cohort of normal subjects before it can be extended to muscle disease states such as muscle dystrophies.

## Figures and Tables

**Figure 1 diagnostics-12-00584-f001:**
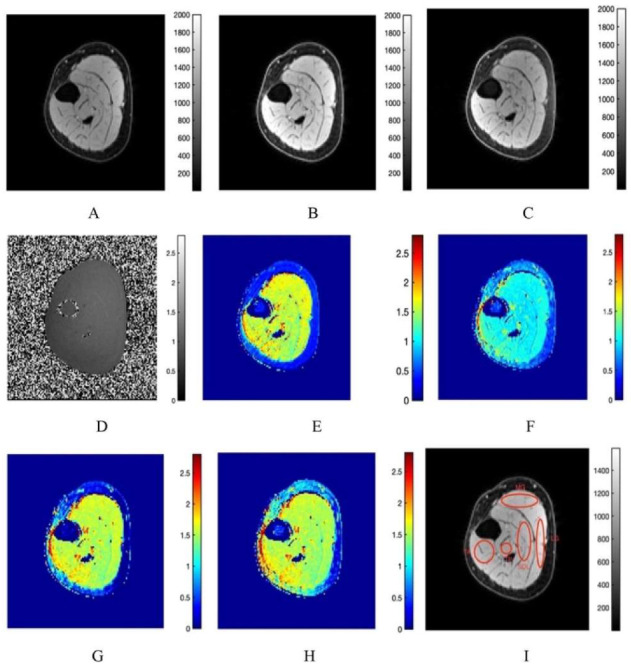
Typical acquired and computed images of the calf muscle (young subject shown here): 5° flip angle (**A**), 15° flip angle (**B**), 25° flip angle (**C**), B1+ map (**D**), T1 map with no fat suppression (**E**), T1 map with fat saturation (**F**), T1 map with water excitation (fast, 1:1 composite pulse) (**G**), T1 map with water excitation (normal, 1:2:1 composite pulse) (**H**). The ROIs in 5 muscle compartments used to extract T1 and MTsat values are shown in (**I**). The legend for the colormap for each computed image is shown next to it. The acquired images are in arbitrary units (**A**–**C**), for the B1+ map unitless (ratio of actual to nominal flip angle) (**D**), and for the T1 maps in seconds (**E**–**H**). It should be noted that images are presented rotated by 90° in order to accommodate all nine images with the color legends in one frame.

**Figure 2 diagnostics-12-00584-f002:**
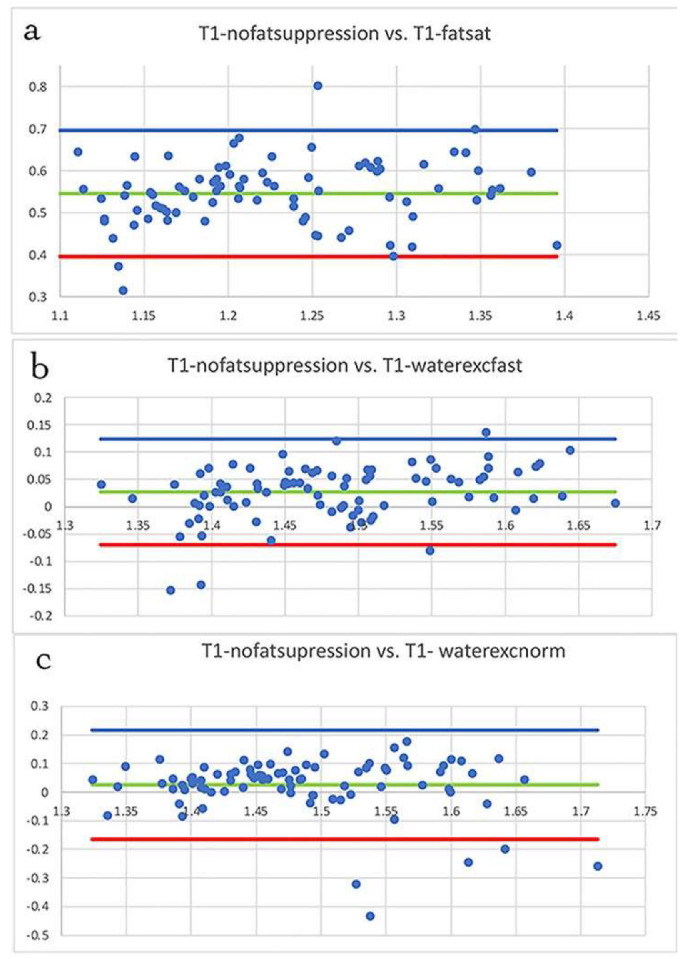
Bland–Altman plot of the difference in T1 values between two measurements (with fat suppression and the reference, without fat suppression) vs. the mean of the two measurements (young subjects only). The *y*-axis is the difference in T1 values between images with and without fat suppression, the *x*-axis is the mean of the two measurements. The data points are T1 values derived from ROIs in each muscle for the ten young subjects. The 95% confidence lines are indicated by the blue and red lines, while the green line is the bias. Using the fat unsuppressed sequence as the reference, BA plots are shown for T1 from the fat saturated sequence (**a**), the water excitation sequence using 1:1 composite pulse (**b**), and the water excitation sequence using 1:2:1 composite pulse (**c**).

**Figure 3 diagnostics-12-00584-f003:**
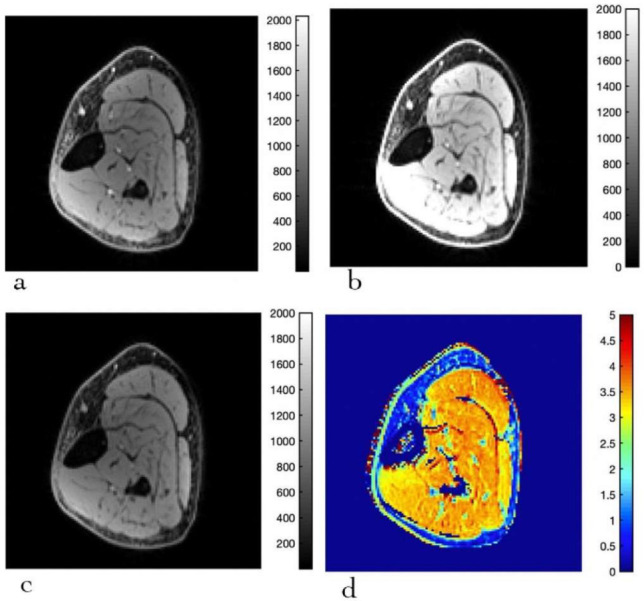
Typical acquired and computed images of the calf muscle for MT_sat_ mapping (senior subject): 10° flip angle with MT pulse (magnetization transfer weighted, MT) (**a**), 20° flip angle (T1-weighted, T1) (**b**), 4° flip angle (proton density weighted, PD) (**c**), MT_sat_ map (**d**). The legend for the colormap for each image is shown next to it. The units for the acquired images are arbitrary units (**a**–**c**), and for the MT_sat_ in % units (**d**).

**Table 1 diagnostics-12-00584-t001:** T1 using no fat suppression and three fat suppression methods: Regional muscle T1 for young and senior cohorts.

T1 in Seconds (Mean ± SD)		TA	MG	LG	SOL	TP
No Fat suppression, T1 (sec) ^b,c,d,e^	Y	1.47 ± 0.1	1.47 ± 0.09	1.59 ± 0.08	1.46 ± 0.04	1.48 ± 0.06
	S	1.53 ± 0.07	1.5 ± 0.11	1.55 ± 0.12	1.44 ± 0.07	1.46 ± 0.06
Fat Saturation, T1 (sec) ^a,b,c,d,e,f,g,h^	Y	1.00 ± 0.07	0.9 ± 0.06	0.98 ± 0.09	0.90 ± 0.04	0.92 ± 0.05
	S	1.04 ± 0.09	0.96 ± 0.0.6	1.01 ± 0.08	0.94 ± 0.07	0.90 ± 0.02
Water Exc.(1:1), T1 (sec) ^a,b,c,d,e^	Y	1.46 ± 0.08	1.44 ± 0.08	1.53 ± 0.08	1.42 ± 0.04	1.44 ± 0.04
	S	1.51 ± 0.06	1.49 ± 0.08	1.57 ± 0.07	1.44 ± 0.04	1.43 ± 0.04
Water Exc.(1:2:1), T1 (sec)^a,b,c,e^	Y	1.45 ± 0.08	1.41 ± 0.08	1.50 ± 0.06	1.41 ± 0.03	1.43 ± 0.04
	S	1.54 ± 0.12	1.52 ± 0.16	1.59 ± 0.09	1.48 ± 0.10	1.45 ± 0.13

Y: Young, S: Senior; medial gastrocnemius (MG), lateral gastrocnemius (LG), sol (SOL), tibialis posterior (TP), tibialis anterior (TA); ^a^ Significant age-related differences, ^b^ Significant difference between LG and MG, ^c^ Significant difference between LG and SOL, ^d^ Significant difference between LG and TA, ^e^ Significant difference between LG and TP, ^f^ Significant difference between TA and TP, ^g^ Significant difference between TA and SOL, ^h^ Significant difference between TA and MG. Values are mean and standard deviation of subjects in each cohort; Significance level: *p* < 0.05.

**Table 2 diagnostics-12-00584-t002:** Regional muscle MT_sat_ values for young and senior cohorts.

MT_sat_ (%) ^a,b,c,d,e^ (Mean ± SD)	TA	MG	LG	SOL	TP
Young (Y)	3.19 ± 0.11	3.47 ± 0.17	3.38 ± 0.16	3.53 ± 0.09	3.41 ± 0.12
Senior (S)	3.16 ± 0.07	3.31 ± 0.15	3.13 ± 0.16	3.36 ± 0.19	3.34 ± 0.18

Medial gastrocnemius (MG), lateral gastrocnemius (LG), sol (SOL), tibialis posterior (TP), tibialis anterior (TA); ^a^ Significant age-related differences (*p* = 0.000039), ^b^ Significant difference between TA and MG (*p* = 0.000146), ^c^ Significant difference between TA and SOL (*p* = 0.000001), ^d^ Significant difference between TA and TP (*p* = 0.001), ^e^ Significant difference between LG and SOL (*p* = 0.003). Values are mean and standard deviation of subjects in each cohort.

## Data Availability

The datasets generated and analyzed during the current study are available from the corresponding author on reasonable request.

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
