# Peer review of "Spin Lattice (T1) and Magnetization Transfer Saturation (MTsat) Imaging to Monitor Age-Related Differences in Skeletal Muscle Tissue"

_diagnostics, 2022, doi:10.3390/diagnostics12030584_

Round 1

Reviewer 1 Report

It is a well structured and written study, with relevant information and solid conclusions. The conclusions are adequately supported by the results presented in the manuscript. I would only suggest minor points to the authors, such as the justification of the sample size and that they review the keywords (they seem too imprecise). Another point that may be interesting in order to support the conclusions of the manuscript is that the authors include all the images as supplementary material as there are few patients in this study.

Reviewer 2 Report

The authors provide a comparison of T1 mapping with MTsat mapping in the identification of regional and age-related muscle changes.

Introduction

line 39-40, please rewrite.

How much longer does a MTsat accquisition usually need?

The inflammatory aspect of aging warrants more detailed description in the introduction. Especially, how this is connected to the fat tissue.

Methods

Subjects Did you assess the necessary power and estimated effect size before recruiting 10 young and 10 old patients?

How did you make sure that the old patients had sarcopenic or dystrophic muscle?

Results

Table 1 and 2: Please be consistent with your selection of muscles / samples in rows or columns. Also, the p-values (i.e. in Table 2) would help the reader to understand better what differences were obtained.

Please depict in the row title what was used (mean/sd/or stderr).

Discussion

The patient cohort should be properly introduced throughout the introduction, since it seems to be a limitation to me that the exact status (regarding diseases, comorbidities, medication) is not mentioned.

The statement "T1 and from MTsat yielding a sensitive biomarker of age-related changes in skeletal muscle". This is not what has been shown in the manuscript. Neither sensitivity nor specificity were or can be assessed in a study with n=20 patients.

The study may give interesting exploratory insights, but no proof-of-concept.

Round 2

Reviewer 2 Report

The authors made substantial and helpful revisions.

The only minor change that is requested is the removal of the word "sensitive" from line 409, since - as mentioned - the sensitivity has not been thoroughly assessed.